# Online Continual Learning for Time Series: a Natural Score-driven Approach

## Abstract

Online continual learning (OCL) methods adapt to changing environments without forgetting past knowledge. Similarly, online time series forecasting (OTSF) is a real-world problem where data evolve in time and success depends on both rapid adaptation and long-term memory. Indeed, time-varying and regime-switching forecasting models have been extensively studied, offering a strong justification for the use of OCL in these settings. Building on recent work that applies OCL to OTSF, this paper aims to strengthen the theoretical and practical connections between time series methods and OCL. First, we reframe neural network optimization as a parameter filtering problem, showing that natural gradient descent is a score-driven method and proving its information-theoretic optimality. Then, we show that using a Student's t likelihood in addition to natural gradient induces a bounded update, which improves robustness to outliers. Finally, we introduce Natural Score-driven Replay (NatSR), which combines our robust optimizer with a replay buffer and a dynamic scale heuristic that improves fast adaptation at regime drifts. Empirical results demonstrate that NatSR achieves stronger forecasting performance than more complex state-of-the-art methods.

## 1 Introduction

Time series forecasting has an impact on both research and the real-world industry. The energy sector (Deb et al., 2017), financial markets (Sezer et al., 2020), macroeconomic policymakers (Clements et al., 2024), and retailing companies (Makridakis et al., 2022), all benefit from accurate predictions. While deep learning had a great impact on the field (Zhou et al., 2021), it is still not unequivocally the best approach for forecasting. On the contrary, it has been shown that in many datasets, simpler statistical methods, such as the class of econometric models with memory (Hamilton, 2020), outperform complex and large neural networks (Godahewa et al., 2021). In macroeconomics, for example, it has been shown that linear models can even outperform nonlinear neural networks (Chi et al., 2025). In addition to this lack of reliable performance, larger models are usually trained in offline batch settings, requiring a full and large training dataset available a priori and assuming no future changes on the relationship between input and output (Sahoo et al., 2018). This is in contrast with a reality where data can be scarce and arrives in streams, with the possibility of experiencing concept drifts in time (Gama et al., 2014).

To create a more realistic and adaptive training setting, it has been proposed to transition to fully online training of the forecaster (Anava et al., 2013). Still, this approach presents multiple challenges for neural networks, like slow convergence (Sahoo et al., 2018), noisy gradients (Bishop & Bishop, 2023), and catastrophic forgetting of previously learned concepts (French, 1999). As with other data structures, learning online from a time series requires both high plasticity to adapt to new regimes and stability to not forget recurrent ones. For this reason, Sahoo et al. (2018) radically reframed online time series forecasting as an online continual learning (Mai et al., 2022) problem.

In this paper, we aim to propose a new second-order algorithm for online learning on nonstationary time series, with strong theoretical backing. In summary:

1. We establish a link between score-driven models (Creal et al., 2013; Harvey, 2013a) and natural gradient descent (Amari, 1998), framing the optimization of deep networks for

    non-stationary forecasting as a filtering task. In particular, we formally prove that natural gradient descent (NGD) is *information-theoretically optimal*.

2. We demonstrate that using a Student's t loss function imposes an upper bound on the update norm. We also propose a novel dynamic adjustment for the scale parameter, which results in improved robustness to outliers, a fundamental property for non-stationary forecasting.

3. Our method **Nat**ural **S**core-driven **R**eplay (**NatSR**), combines these theoretical advances and achieves state-of-the-art performance in OCL, outperforming existing methods on 5 out of 7 datasets.

## 2 BACKGROUND

Online time series forecasting follows the online learning paradigm (Shalev-Shwartz et al., 2012): at each time step, a model makes a prediction and, after observing reality, it is adjusted using that information. Online time series forecasting applies this to time series data, observing the data in time order, and updating the model with each new observation. Let $\{x_t\}_{t \in \mathbb{Z}}$ denote the input time series and $\{y_t\}_{t \in \mathbb{Z}}$ the corresponding target time series, for which $x_t \in \mathbb{R}^s$ with $s \in \mathbb{N}$ and $y_t \in \mathbb{R}^d$ with $d \in \mathbb{N}$, for any $t \in \mathbb{Z}$. As usual, each time series is regarded as a realization of an underlying stochastic process. Specifically, let $\{X_t\}_{t \in \mathbb{Z}}$ denote the input process and $\{Y_t\}_{t \in \mathbb{Z}}$ the output process generating the observed series. We consider the filtration $\mathscr{F} = \{\mathscr{F}_t\}_{t \in \mathbb{N}}$ where $\mathscr{F}_t = \sigma(X_{1:t}, Y_{1:t})$, so that $\mathscr{F}_t$ contains all information available up to time $t$. Here we use the shorthand notation $X_{1:t} = \{X_1, \cdots, X_t\}$. Given the input $x_t$ and the network weights $w_t \in W \subset \mathbb{R}^d$, the network produces the output $\theta_t(w_t) = f_{w_t}(x_t) \in \mathbb{R}^d$. With standard gradient descent, the weights are updated as $w_{t+1} = w_t + \eta \nabla_{w_t} \mathscr{L}(y_t, \theta_t)$, where $\mathscr{L}$ is a loss function.

This process aims to learn the right weights for the current time, in particular, changing them when the data regime changes. Unfortunately, this can result in catastrophic forgetting (French, 1999) with the model losing previous knowledge of past regimes. Continual Learning (Lange et al., 2022) is a field that tries to solve this, making models able to accumulate knowledge consistently in non-stationary environments. More specifically, Online Continual Learning (OCL) does so by accessing each observation a single time in an online learning fashion. Hence, it requires the method to have a balance between fast adaptation and stability, without knowing when a regime change happens. Such an approach is similar to human learning. The additional complexity of applying OCL to time series is that both virtual and real drifts can happen (Gama et al., 2014): in virtual drifts, new portions of the input space are explored, while in real drifts, the relation between input and output changes. This requires an even more complex balance between plasticity and stability.

## 3 RELATED WORKS

**Online Time Series Forecasting:** In recent years, more and more works have explored the use of deep learning for time series forecasting, proposing a variety of specialized architectures (Salinas et al., 2020; van den Oord et al., 2016; Bai et al., 2018; Zhou et al., 2021). Unfortunately, they are not directly applicable to online time series forecasting (Anava et al., 2013) due to concept drift (Gama et al., 2014). Fekri et al. (2021) showed that an online RNN achieves stronger results than standard online algorithms or offline trained neural networks for energy data. Wang et al. (2021) proposed IncLSTM, fusing ensemble learning and transfer learning to update an LSTM incrementally. Naive online time series forecasting can suffer from forgetting (French, 1999) in non-stationary streams (Sahoo et al., 2018; Aljundi et al., 2019).

**Online Continual Learning (OCL):** Most OCL methods in the literature use replay to mitigate forgetting (Soutif-Cormerais et al., 2023). However, (Soutif-Cormerais et al., 2023) showed that SOTA approaches still can have more forgetting than a simple replay baseline (Aljundi et al., 2019). Recent works highlighted a "stability gap" (Caccia et al., 2022; Lange et al., 2023), where the model suddenly forgets at task boundaries. Relevant to this work, multiple optimization-based approaches constrain the update to remove interference. GEM (Chaudhry et al., 2019a; Lopez-Paz & Ranzato, 2017) enforce non-negative dot product between task gradients, while other use orthogonal projections (Saha et al., 2021; Farajtabar et al., 2020). For OCL, it has been shown how a combination of GEM and replay can mitigate the stability gap (Hess et al., 2023). More recently, LPR (Yoo

et al., 2024) proposed an optimization approach for OCL, combining replay with a proximal point method. Improving on that, OCAR (Urettini & Carta, 2025) proposed the use of second-order information. (Abuduweili & Liu, 2023) showed the efficacy of feedforward adaptation when compared with feedback adaptation OCL methods.

**Online Continual Learning and Forecasting:** With modern deep models, multiple time series regimes can be learned within a single network. Sahoo et al. (2018) propose reframing online time series forecasting as task-free OCL, removing the need for manual labeling of task boundaries. FSNET (Pham et al., 2023) maintains a layer-wise EMA of the gradients to adapt the weights to the current tasks via a hypernetwork. OneNet (Wen et al., 2023) keeps two separate neural networks to model cross-variate relationships and temporal dependencies separately, combining the two separate forecasts dynamically using offline reinforcement learning. Very recently, Zhao & Shen (2025) proposed PROCEED to solve the delay caused by the time needed for the realization of the whole prediction length to happen before making the update.

## 4 NATURAL SCORE-DRIVEN REPLAY

The main aim of this part of the paper is to show the information-theoretic optimality of Natural Gradient Descent (NGD) and to describe the main building blocks of NatSR. First, in section 4.1, we show that the NGD can be interpreted as a score-driven update. In section 4.2 we then prove its information-theoretic optimality. Then, in section 4.3, we show that the NGD used with a Student's t distributional assumption enforces a bounded update. Finally, by adding both memorization (section 4.4) and fast-adaptation mechanisms (section 4.5), we obtain NatSR.

### 4.1 FROM SCORE-DRIVEN MODELS TO NATURAL GRADIENT DESCENT

Score-driven models, known both as Generalized Autoregressive Score (GAS) models (Creal et al., 2013) and Dynamic Conditional Score Models (DCS) (Harvey, 2013b), are approaches to estimate time-varying parameters from a time series. They are observation-driven models, following the categorization by Cox (1981). In this framework, the dynamics of the time-varying parameter vector are governed by the score of the conditional likelihood function of the observed variable. The score is defined as the gradient of the log-likelihood with respect to the parameters. Score-driven models specify the evolution of time-varying parameters as an autoregressive process Hamilton (2020) with an innovation term driven by the score. In essence, following the score provides a data-driven update that moves the parameters in the direction that increases the likelihood of the observed data, in a process similar to gradient descent optimization.

Formally, $y_t \sim p(y_t \mid \theta_t, \phi)$ where $\phi = (\omega, A_1, \cdots, A_m, B_1, \cdots, B_n)$ denotes the static parameters, and the time-varying parameter $\theta_t$ evolves according to

$$\theta_{t+1} = \omega + \sum_{i=1}^{m} A_i \, s_{t-i+1} + \sum_{j=1}^{n} B_j \, \theta_{t-j+1}, \tag{1}$$

with $s_t = S_t \nabla_t$. Here, $\nabla_t = \frac{\partial \log p(y_t|\theta_t,\phi)}{\partial \theta_t}$ denotes the score of the conditional log-likelihood with respect to $\theta_t$ and $S_t$ is a scaling matrix. The original proposers of GAS models suggested the use of the Fisher Information Matrix (FIM) for $S_t$.

This connects us to natural gradient descent, a fundamental optimization algorithm used in machine learning (Amari, 1998), which we can interpret as a special case of a score-driven model under suitable conditions. In natural gradient descent, the weights are updated as

$$w_{t+1} = w_t + \eta \, \mathscr{I}_t^{-1}(w_t) \nabla_{w_t}(y_t), \tag{2}$$

where $\eta \in \mathbb{R}$ is the constant learning rate, $\mathscr{I}_t \in \mathbb{R}^{d \times d}$ is the FIM and $\nabla_{w_t}(y_t) = \frac{\partial \log p(y_t|\theta_t)}{\partial \theta_t} \frac{\partial \theta_t}{\partial w_t} \in \mathbb{R}^d$ is the score, i.e. the gradient of the log-likelihood function, while $\theta_t(w_t) = f_{w_t}(x_t)$ corresponds to the provisional output of the network before the update. Thus, the time-varying parameter update of the score-driven model (see Eq.(1)) reduces to the natural gradient descent for $m = n = 1, \omega = 0, A_1 = \eta, B_1 = 1$ and $S_t = \mathscr{I}_t^{-1}$. The result of this is that NGD can be interpreted as a filtering mechanism that, with each new observation, updates the estimation of a neural network's weights, following

the score to maximize the likelihood function. Unlike standard gradient descent, which assumes all directions in parameter space are equally meaningful, the natural gradient incorporates the Fisher information matrix to rescale the gradient. Hence, the filtering process takes into consideration the curvature of the parameter space: which directions have higher/lower gradient variance. All of this makes NGD equivalent to a score-driven model, hence capable of estimating the weights in a nonstationary environment. Viewing NGD as a statistical filter has already been proposed by Ollivier (2018) under a different framework, where the equivalence between the Kalman filter and the NGD was shown.

## 4.2 INFORMATION-THEORETIC OPTIMALITY

The main result of this section is Proposition 4.1, where we show that the expected updated weight always lies closer to the optimal weight vector than the weight before the update.

After the update (see Eq.(2)) the output is $\theta_t(w_{t+1}) = f_{w_{t+1}}(x_t)$. This output can be interpreted as the parameter vector of an assumed density when the loss function is derived directly from a specific likelihood function. For example, minimizing the mean-squared error (MSE) is equivalent to performing maximum likelihood estimation under the assumption of normally distributed errors (Bishop, 2006). We thus postulate a statistical model:

$$y_{t+1} \mid \mathcal{F}_t \sim p_{t+1|t+1} := p(\cdot \mid \theta_t(w_{t+1})) \tag{3}$$

which approximates the true conditional density of the target time series, i.e. $y_{t+1} \mid \mathcal{F}_t \sim q_{t+1}$ and $p_{t+1|t} := p(\cdot \mid \theta_t(w_t))$ is the statistical model implied by the weights before the update.

We show that the weight update obtained via natural gradient descent (see Eq.(2)) reduces the KL divergence between the assumed model and the true statistical model, relative to the divergence before the update. In particular, we demonstrate that the parameter update from $w_t$ to $w_{t+1}$ moves, in expectation, closer to the weight vector $w_t^*$ which corresponds to the pseudo-true time-varying parameter $\theta_t^*$, that is defined as

$$\theta_t^* = \underset{\theta \in \Theta}{\arg\min} \underbrace{\int_{\mathbb{R}^d} q_t(y) \log \frac{q_t(y)}{p(y|\theta)} dy}_{\mathrm{KL}_t(\theta)} = \underset{\theta \in \Theta}{\arg\max} \, \mathbb{E}_{y \sim q_t}[\log p(y|\theta)], \tag{4}$$

Hence it is the value that minimizes the KL divergence between the postulated and the true statistical model. Consequently, neural networks trained with the natural gradient can be regarded as information-theoretically optimal, in the sense of Blasques et al. (2015); Gorgi et al. (2024).

We introduce the following assumptions:

**(A1)** Assume that there exists $w_t^* \in W$ such that $\theta_t^* = f_{w_t^*}(x_t)$.

For the second assumption we define the function $g_t : W \to \mathbb{R}$ such that $g_t(w) = \mathbb{E}_{t-1}[\log p(y_t|w)]$ where $\mathbb{E}_t[\cdot] = \mathbb{E}[\cdot \mid \mathcal{F}_t]$. Thus, the function $g_t$ corresponds to the expected log-likelihood at time $t$ given the information up to time $t-1$.

**(A2)** Assume that $g_t(w) \in C^2(W)$ with $W$ open and convex and

$$\nabla g_t(w) = \mathbb{E}_{t-1} \nabla_w(y_t) = \mathbb{E}_{t-1} \frac{\partial \log p(y_t|\theta)}{\partial \theta} \frac{\partial \theta}{\partial w}$$

where $\theta = f(w)$ and $\frac{\partial \theta}{\partial w}$ denotes the Jacobian matrix whose $(i,j)$ entry is $\frac{\partial \theta_i}{\partial w_j}$.

In $(A2)$ we assume that $g_t(\cdot)$ is twice differentiable and that we can interchange the derivative with the expectation.

**(A3)** For any $w_1, w_2 \in W$, there exists $c > 0$ such that:

$$\langle \mathscr{I}_t(w_1)^{-1} \nabla g_t(w_1) - \mathscr{I}_t(w_2)^{-1} \nabla g_t(w_2), w_1 - w_2 \rangle \le$$

$$-\frac{1}{c} \| \mathscr{I}_t(w_1)^{-1} \nabla g_t(w_1) - \mathscr{I}_t(w_2)^{-1} \nabla g_t(w_2) \|^2, \quad \langle \cdot, \cdot \rangle \text{ is the inner product on } \mathbb{R}^d$$

Under assumption (A3), the expected score innovation is a decreasing Lipschitz continuous function.

The weights with the natural gradient are updated as in Eq.(2), the expected weight update parameter given the information $\mathscr{F}_{t-1}$ is then

$$\mathbb{E}_{t-1}[w_{t+1}] = w_t + \eta \mathscr{I}_t^{-1}(w_t)\mathbb{E}_{t-1}[\nabla_{w_t}(y_t)].$$

**Proposition 4.1.** *Let assumptions* (A1)-(A3) *hold with* $0 < \eta < \frac{2}{c}$, *then*

$$\|\mathbb{E}_{t-1}[w_{t+1}] - w_t^*\| < \|w_t - w_t^*\|.$$

Moreover, assuming that the network output is locally Bi-Lipschitz on the weights in a neighborhood of $w_t^*$ (**A4**), we can derive the corresponding theoretical optimality result that transfers from the weight space to the output space. The assumption of bi-Lipschitzness is reasonable, as there exist neural networks that satisfy it even globally (see Wang et al. (2024)). A bi-Lipschitz neural network can jointly and smoothly control its Lipschitz continuity—its sensitivity to input perturbations—and its inverse Lipschitz property, which preserves meaningful separation between inputs that yield different outputs.

**Proposition 4.2.** *Let assumptions* (A1)-(A4) *hold with* $0 < \eta < \frac{2}{c}$, *then*

$$\|\theta_t(\mathbb{E}_{t-1}[w_{t+1}]) - \theta_t^*\| < \|\theta_t(w_t) - \theta_t^*\|.$$

The proofs can be found in Appendix A.2.

### 4.3 ENFORCING A BOUNDED UPDATE

Outliers are detrimental to methods that filter parameters at each observation. For this reason, robust score-driven models use bounded scores derived from heavy-tailed distributions (like the Student's t) (Artemova et al., 2022). Controlling the update norm is one of the main characteristics of successful optimizers like ADAM (Kingma & Ba, 2015).

**Theorem 4.1.** *Let the loss function be the one induced from a Student's-$t_\nu(s)$ distribution:*

$$\underbrace{-\log p(y \mid f(x))}_{loss} = -\log\left(\frac{\Gamma(\frac{\nu+1}{2})}{\Gamma(\frac{\nu}{2})\sqrt{\pi\nu}}\right) + \frac{1}{2}\log(s^2) + \frac{\nu+1}{2}\log\left(1 + \frac{(y-f(x))^2}{\nu s^2}\right),$$

*then using the Tikhonov regularization, the following bound holds:*

$$\|\tilde{\nabla}_w \log p(y|f_w(x))\|_2 \le \frac{1}{4}\sqrt{\frac{(\nu+1)(\nu+3)m}{\tau\nu}}. \tag{5}$$

*where $m$ is the number of outputs and $\tau$ the Tikhonov regularization constant.*

The proof can be found in Appendix A.3.

Assuming the target time series follows a Student's $t_\nu$ distribution induces a specific loss for the natural gradient update that is inherently bounded. Intuitively, the FIM provides a bound on the Jacobian, as it involves the product of the Jacobian and its transpose (see Appendix A.3 for more details). At the same time, the Student's $t_\nu$ distribution bounds the gradient of the loss with respect to the outputs (see Figure 2). As a result, the full natural gradient with a Student's $t_\nu$ negative log-likelihood loss has a bounded $L_2$-norm, making the optimization process more robust to outliers. Figure 1a shows the effects of bounded updates on a toy example of a noisy sinusoid with outliers. OGD's gradients spike in response to outliers, destabilizing the optimization, whereas NatSR's natural gradient with a Student's $t_\nu$ loss remains bounded, yielding stable and robust updates.

### 4.4 MEMORY BUFFER

We showed that the natural gradient update shares the same information-theoretic optimality with score-driven models and that the use of the Student's t log-likelihood can bound the update. Now we add to this filtering method the ability to accumulate knowledge without catastrophic forgetting. We

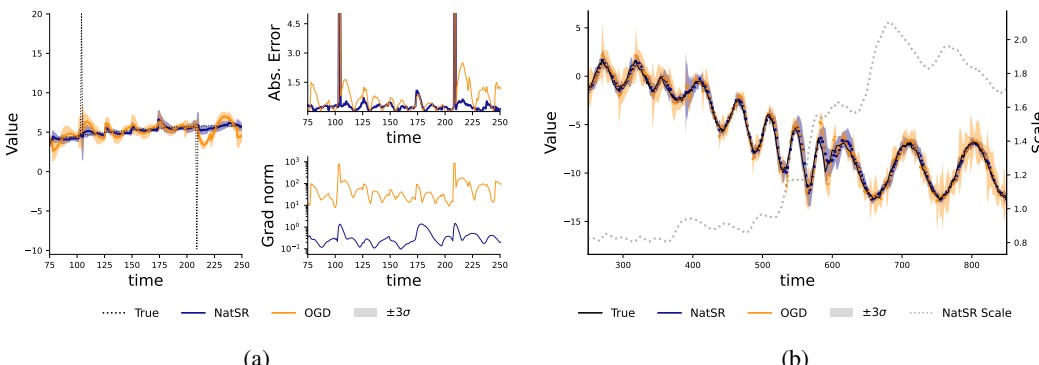

(a)                                                        (b)

Figure 1: Mean predictions and standard deviations of NatSR and simple Online Gradient Descent (OGD) on a noisy sinusoidal wave under two challenging conditions: (left) with outliers and (right) with changing regimes, each repeated ten times. In the outlier setting (a), OGD is destabilized and requires several iterations to recover accurate forecasts, whereas NatSR remains stable. The bottom-right panel in (a) highlights the difference in update magnitudes: OGD's gradients grow by an order of magnitude in response to the outlier, while NatSR's remain comparable to those from normal errors. In the regime-change setting (b), the scale rises during transitions, allowing for larger gradients and faster updates, and decreases again once the series stabilizes. This dynamic scaling, combined with second-order information from the FIM, enables NatSR to adapt rapidly to changes in both amplitude and frequency, as reflected by the smaller standard deviations during the second regime compared to OGD.

use a simple Experience Replay approach (Chaudhry et al., 2019b) with a second-order approximation (Urettini & Carta, 2025) to recover a natural gradient update. At each step in time, the optimal second-order update is the solution of the problem

$$\min_{\delta} \quad \nabla_{N_t}^T \delta + \frac{1}{2}\delta^T \mathbf{H}_{N_t}\delta + \lambda\nabla_{B_t}^T\delta + \frac{\lambda}{2}\delta^T\mathbf{H}_{B_t}\delta \quad \text{subject to} \quad \frac{1}{2}||\delta||_2^2 \leq \varepsilon, \qquad (6)$$

which is solved by

$$\delta_t^* = -(\mathbf{H}_{N_t} + \lambda\mathbf{H}_{B_t} + \tau\mathbf{I})^{-1}(\nabla_{N_t} + \lambda\nabla_{B_t}),$$

where $\delta_t^*$ is the optimal optimization step given the information at time $t$, $N_t$ are the new observations done at time $t$, $B_t$ is a set of observations sampled from the buffer $\mathscr{B}$, $\tau$ is the Tikhonov regularization, $\lambda$ the importance given to the past, $\mathbf{H}$ the Hessian matrix and $\nabla$ the gradient vector.

Following Urettini & Carta (2025), we note that the Fisher Information matrix $\mathscr{I}$ is a Generalized Gauss-Newton matrix that approximates the Hessian (Martens, 2020). Hence we get:

$$\delta_t^* = -(\mathscr{I}_{N_t} + \lambda\mathscr{I}_{B_t} + \tau\mathbf{I})^{-1}(\nabla_{N_t} + \lambda\nabla_{B_t}). \qquad (7)$$

To improve optimization speed (Yuan et al., 2016; Sutskever et al., 2013) and reliability with noisy data, such as time series data, we take inspiration from ADAM (Kingma & Ba, 2015) and smooth the natural gradient update with an EMA:

$$\delta_t^{EMA} = \alpha^{EMA}\delta_t^* + (1 - \alpha^{EMA})\delta_t^{EMA}. \qquad (8)$$

## 4.5 Dynamic Scale

The Student's t decreases the score for larger errors after a certain threshold that depends on the degrees of freedom (see Figure 2). Unfortunately, this approach may result in slow updates during sudden regime changes due to the small score. This would be in contrast with the *fast adaptation* desiderata of OCL.

To address this, we propose to adjust the step size dynamically. First, notice that the natural score of the Student's t mean converges to the (unbounded) Gaussian score when we increase the scale parameter $s$ (see Figure 2) of the Student's t likelihood 4.1. When a new regime occurs, the model error will increase substantially, and with it, the observed variance of the target conditional on our predicted means. By also increasing $s$ to follow the observed variance, the step size increases with it.

Unfortunately, this step size is not directly controlled by $s$ when the Tikhonov regularization is used (like in our case). As a matter of fact, the update bound we found in equation 5 does not depend on $s$. To recover the same effect as in score-driven models, we propose to set the Tikhonov regularization as $\tau = \frac{0.9\beta}{1+s^2} + \frac{0.1\beta}{s^2}$ (see Appendix B for the derivation) where $\beta$ is a scalar hyperparameter. The result is that an increase in $s$ would increase the gradient bound (Eq. 5) through the decrease of $\tau$.

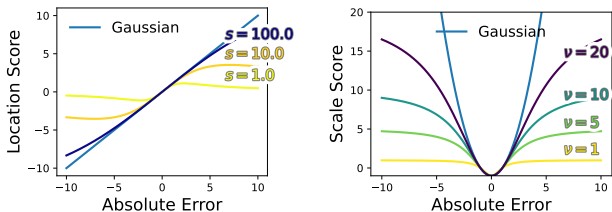

Figure 2: Natural score of the Student's t compared to a Gaussian score. *Left*: score of the mean for different scales. *Right*: score of the scale parameter for different $\nu$.

To maintain robustness against outliers, we need to update $s$ gradually with bounded updates, so that only multiple consecutive unexpected observations would significantly increase the bound. We propose to use once again the score-driven update strategy, deriving the score of the scale from the same log-likelihood used as objective for our model $f_w(x)$. The score-driven update using the score of a Student t log-likelihood related to $s^2$ regularized by its relative Fisher information is (Artemova et al., 2022):

$$s_{t+1}^2 = s_t^2 + \alpha_s \frac{s_t^2 \nu (e_t^2 - s_t^2)}{s_t^2 \nu + e_t^2},$$

where $e_t = y_t - f_w(X_t)$ and $\alpha_s$ is a learning rate. Additionally, we enforce a lower bound on $s_{t+1}^2$ to avoid values too small. In Figure 2, the regularized score for the scale is visualized. The effect of this dynamic scale is as follows: when the squared error is larger than $s_t^2$, the variance is larger than expected, and the scale $s_t^2$ starts to adjust, with a speed controlled by the degrees of freedom $\nu$ and the parameter $\alpha_s$. If the observed error is an outlier, the effect is limited to this single bounded update of the scale. If instead the squared error remains larger than $s_t^2$, it is interpreted as a regime shift and $s_t^2$ continues to increase. The increase of $s_t^2$ directly influences the natural gradient (see Appendix B), and increases the upper bound of the update, allowing the model to adapt faster. On the other hand, when the predictions of the model are accurate, the scale is lowered, decreasing the bound and making the model more stable. An example of the possible effects of the dynamic scale is shown in Figure 1b, where the value of the scale increases as the regime of the data changes, allowing for larger updates and faster adaptation.

To sum up, we showed that natural gradient, besides its well-known properties as an optimizer (Amari, 1998; Martens, 2020; Kunstner et al., 2019), can also be interpreted as a score-driven model, sharing the same information-theoretic optimality for nonstationary data. When we combine a Student's t negative log-likelihood loss function, the weight update is bounded and robust to outliers. Adding a memory buffer to this allows the model to "remember" also past regimes, accumulating knowledge in time. Finally, with the dynamic score, we enable both stability and fast adaptation. All of this is the **Nat**ural **S**core-Driven **R**eplay (NatSR). The full algorithm can be found in Appendix E and some additional implementation details in Appendix C.

## 5 EXPERIMENTS

We empirically validate our proposal following the setup in Pham et al. (2023). An extended discussion of the experimental setup is provided in Appendix F. Our full repository used for the experiments can be found at `https://anonymous.4open.science/r/NatSR`.

**Baselines:** We compare our method against state-of-the-art methods such as Experience Replay (ER) (Chaudhry et al., 2019b), DER++ (Buzzega et al., 2020), FSNET (Pham et al., 2023), and OneNet (Wen et al., 2023). We also tested a simple online gradient descent approach (OGD), where the target is to adapt to newly observed data, with no memorization objective.

**Datasets:** We test on the same real-world datasets as FSNET, covering a wide range of sources and behaviours. The ETT dataset (Zhou et al., 2021) collects the oil temperature and other 6 power load features from different transformers with hourly ("h") or 15-minute ("m") frequency. ECL[1]

---

[1]`https://archive.ics.uci.edu/ml/datasets/ElectricityLoadDiagrams20112014`

| Dataset | Pred. Len | OGD | ER | DER++ | FSNET | OneNet | NatSR (Ours) |
|---|---|---|---|---|---|---|---|
| ECL | 1 | 1.67 | 1.43 | 1.11 | 0.96 | **0.64** | 3.53 |
| | 24 | 3.12 | 3.21 | 2.89 | 1.42 | **0.92** | 4.01 |
| | 48 | 3.27 | 3.01 | 2.96 | 1.44 | **0.96** | 4.14 |
| ETTh1 | 1 | 0.87 | 0.83 | 0.82 | 0.92 | 0.83 | **0.79** |
| | 24 | 1.50 | 1.49 | 1.45 | 1.08 | 1.38 | **0.97** |
| | 48 | 1.46 | 1.42 | 1.42 | 1.16 | 1.39 | **1.12** |
| ETTh2 | 1 | 1.14 | 1.09 | 1.08 | 1.10 | 1.06 | **1.01** |
| | 24 | 1.65 | 1.60 | 1.60 | 1.37 | 1.56 | **1.23** |
| | 48 | 1.63 | 1.62 | 1.61 | 1.48 | 1.61 | **1.39** |
| ETTm1 | 1 | 1.18 | 1.03 | 1.01 | 1.10 | 1.05 | **0.97** |
| | 24 | 2.19 | 1.82 | 1.83 | 1.45 | 1.91 | **1.16** |
| | 48 | 2.19 | 1.91 | 1.81 | 1.52 | 2.04 | **1.33** |
| ETTm2 | 1 | 1.36 | 1.26 | 1.24 | 1.14 | 1.15 | **1.12** |
| | 24 | 2.03 | 1.80 | 1.81 | 1.48 | 1.79 | **1.37** |
| | 48 | 2.01 | 1.85 | 1.82 | 1.51 | 1.84 | **1.50** |
| Traffic | 1 | 0.84 | 0.79 | 0.78 | 0.70 | **0.62** | 0.89 |
| | 24 | 1.05 | 1.07 | 0.95 | 0.96 | **0.91** | 1.14 |
| WTH | 1 | 1.47 | 1.30 | 1.44 | 1.19 | 1.06 | **1.04** |
| | 24 | 1.98 | 1.90 | 1.84 | 1.44 | 1.73 | **1.25** |
| | 48 | 1.97 | 1.89 | 1.86 | 1.46 | 1.82 | **1.39** |

Table 1: Average MASE across 3 runs. Best in **bold**, second best underlined.

represents the electricity consumption of 321 clients from 2012 to 2014. WTH[2] is a collection of weather features from multiple locations in the US. Traffic[3] measures the traffic on the San Francisco Bay Area freeways.

**Experimental Procedure:** All methods undergo an offline warm-up phase using the first 20% of the data for training, and the following 5% for validation and early stopping. This phase is always done using AdamW (Loshchilov & Hutter, 2017) with a learning rate schedule. The remaining 75% of the data is used for online training and evaluation, with model updates at each new observation. During the online phase, the optimizer is reset and possibly changed, using a different learning rate. The optimal value of this online learning rate is selected with a hyperparameter tuning in a full online training with the ETTh1 dataset. The tuning of the online learning rate is the only difference with the FSNET approach. We believe that without a transparent tuning of this parameter, it is very hard to compare different methods. Following the guidelines of Godahewa et al. (2021), we use MASE (Hyndman & Koehler, 2006) as our main evaluation metric.

## 5.1 RESULTS

In Table 1, we report the average MASE over three runs with different random seeds. Appendix G explores additional configurations of NatSR, FSNET, and OneNet, such as FSNET and OneNet with MSE losses and a less conservative version of NatSR with $\nu = 500$ and AdamW optimizer. The configurations shown in Table 1 are the best for each method. We also report the standard deviations and training times of the online phase in Appendix G. Figure 3, instead, gives a more qualitative analysis of our method, compared with our baselines in different scenarios.

NatSR obtained the best MASE on 5 of 7 datasets. It is interesting to note that whenever NatSR is the best method, FSNET follows as second-best, suggesting that whenever continual learning is fundamental, CL-focused solutions are necessary, with NatSR being the better solution. Notice that NatSR achieves these results by only changing the loss and the optimizer without any architectural solution customized for time series like FSNET and OneNet. Unfortunately, we notice that on two datasets, ECL and Traffic, NatSR reaches a higher MASE compared to more complex methods. We noticed that these two datasets have the highest number of features, and are the only ones where

---

[2]https://www.ncei.noaa.gov/data/local-climatological-data/
[3]https://pems.dot.ca.gov/

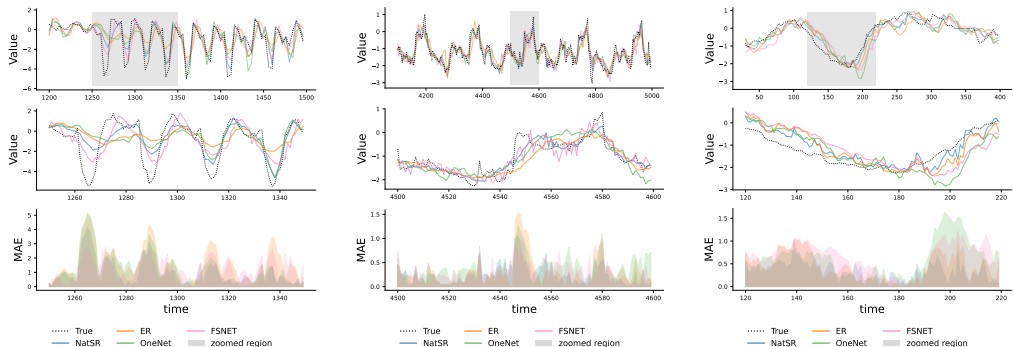

Figure 3: Forecasting results on three datasets: (left) ETTh1 demonstrates the model's ability to adapt quickly; (middle) ETTm1 illustrates the stability of our model, producing less noisy predictions compared to baselines such as FSNET; (right) WTH highlights the importance of replay, as NatSR and ER achieve the best performance when revisiting previously observed input ranges.

| Scale | Replay | ETTh1 | | ETTm1 | | WTH | |
|---|---|---|---|---|---|---|---|
| | | MASE | Rel. Δ | MASE | Rel. Δ | MASE | Rel. Δ |
| ✓ | ✓ | 0.97 | - | 1.16 | - | 1.25 | - |
| ✓ | - | 1.10 | -13% | 1.29 | -11% | 1.35 | -8% |
| - | ✓ | 1.02 | -5% | 1.22 | -5% | 1.32 | -6% |
| - | - | 1.15 | -19% | 1.34 | -16% | 1.40 | -12% |

Table 2: MASE of NatSR with 50 degrees of freedom and its variants for prediction length 24.

OGD performs better than ER in at least a prediction length, suggesting the possibility that these datasets require more plasticity than stability.

Besides the results of NatSR, these experiments confirmed once again the potential of online continual learning approaches to improve online time series forecasting. Standard OGD is rarely able to overcome ER, and more sophisticated OCL methods perform much better. Each method is evaluated on its online forecasting ability and on streams of data that are not a synthetic simulation of a task or domain-switching setting. These are real data, actually observed in a specific time order, that can suffer real or virtual drifts naturally. Still, OCL methods show large improvements when compared to standard online gradient descent learning, underlying the importance of learning stability in OTSF.

## 5.2 ABLATION STUDY

To better understand the role of each component of NatSR, we conduct an ablation study by selectively removing two key mechanisms: the replay strategy and the dynamic scale. Table 2 reports the MASE of each variant, along with the relative performance loss compared to the original version of our method. All results are obtained with 50 degrees of freedom for the Student's t-distribution and a forecasting horizon of 24.

The results clearly indicate that both components are beneficial, although their contributions differ in strength. Removing the replay buffer leads to drops in performance between 8% and 13%, while the dynamic scale causes smaller but consistent losses of about 5-6%. The importance of replay is in line with expectations, as Experience Replay improves substantially compared to online gradient descent in our experiments. Interestingly, however, the relative gain from adding replay within our method is even larger than the gain obtained by simply equipping SGD with replay. This suggests a synergistic effect: replay not only provides access to past samples, but also interacts favorably with our second-order optimization scheme.

A notable observation arises when both mechanisms are removed: the resulting degradation, up to 19%, is equal to or larger than what one might predict from the sum of the individual effects. This suggests that our method is able to leverage replay and dynamic scaling in a complementary

way: replay provides stability across tasks, while scaling enhances adaptability. Their joint effect is greater than the sum of the parts, indicating that the full method is particularly effective at handling non-stationary data streams.

## 5.3 DISCUSSION AND LIMITATIONS

With NatSR, we introduced a method that is rooted in score-driven models and natural gradient descent. The use of the Fisher Information together with Experience Replay has the double effect of accelerating learning (Amari, 1998) while penalizing directions that are important for past data. While we showed the empirical potential on standard end-to-end training, future works can explore the extension of NatSR to the finetuning of foundation models, where keeping intact previous knowledge, while also learning efficiently from scarce data, is of primary importance.

In our proposal, the use of the Student's t is fundamental to obtaining a bounded update, something that can be crucial in datasets where sudden changes and outliers can disrupt learning. On the other hand, even when using the dynamic scale to adjust the bound, some datasets require much less stability and more plasticity. ECL and Traffic are examples of this. They both present large, sudden, and persistent regime changes, where memory and stability are not rewarded. The robustness of NatSR, while very useful in the other datasets, still results in updates that are too conservative for the fast changes in ECL and Traffic, causing a larger error. As a preliminary step towards a possible solution, we notice that if we increase the degrees of freedom and use ADAM (Kingma & Ba, 2015) on top of our method, we obtain a version of NatSR that is much stronger on ECL and Traffic, but weaker on the other datasets (App. F). Designing a single method that provides both fast adaptation and robustness to forgetting is still an open challenge. Time series forecasting is particularly complex in this regard, as different datasets can require widely different approaches. Moreover, it is possible that the hyperparameter tuning done at the beginning will become invalid in the future due to sudden regime changes. Exploring adaptive hyperparameters in the same fashion as our dynamic scale could become fundamental to advancing online time series forecasting.

## 6 CONCLUSION

In this paper, combining theoretical and empirical insights from online continual learning and econometrics, we proposed NatSR, a novel method for online time series forecasting. We proved a formal connection between score-driven models and natural gradient descent, showing its information-theoretic optimality. We also proved that the combination of natural gradient and Student's t loss yields a bound on the update, making the learning more robust. Then, we introduced a dynamic scale of the Student's t to adapt online the plasticity of the model. Building on these insights, we proposed NatSR as a combination of natural gradient, Student's t loss, memory buffer, and dynamic scale. Empirical results show competitive performance against state-of-the-art methods, showing the potential of developing new OCL methods starting from time series analysis theory. Overall, OTSF provides a challenging and realistic application scenario for continual learning methods, where the balance between stability and plasticity is dataset-dependent and may change over time. The open question is whether this trade-off can be adjusted automatically to have a single robust method for every dataset.

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

# A  DERIVATIONS OF THE THEORETICAL RESULTS

In this section, we demonstrate the theoretical results by first reviewing some facts for the GAS model and highlighting its similarities with the neural networks when the optimization is the natural gradient (section A.1). Then in section A.2 we state the proof of the propositions for the optimality of the parameters and finally in section A.3 we demonstrate the update bound.

## A.1  A NOTE ON GAS MODEL

Setting $S_t = \mathbf{I}$ in the GAS model would make the filtering of $w_t$ equivalent to SGD. More interestingly, Creal et al. (2013) suggested the use of the Fisher information matrix as the rescaling matrix $S_t$. This would make the GAS update of $w_t$ equivalent to a natural gradient descent (Amari, 1998). We are not the first ones to draw a connection between natural gradient descent and time series filtering. Ollivier (2018) already showed formally that natural gradient descent can be cast as a special case of the Kalman filter. With GAS models, the connection is more straightforward, as it directly derives from the definition of the GAS update itself by considering as time-varying parameters the weights of the network and not the likelihood parameters themselves. Hence, we can interpret the online optimization process not as a way to find a static optimum as more data arrives, but as a way to respond to new information, filtering the values of the weights and finding the best way to "follow" a changing loss landscape. Following the suggestions of Creal et al. (2013), and the results of Ollivier (2018), we suggest adapting to new observations using the log-likelihood score regularized by the inverse FIM. Moreover, GAS models have been widely used with high-kurtosis distributions like the Student's-t distribution, gaining robustness to outliers (Artemova et al., 2022). We show that the combination of the inverse FIM gradient preconditioning and of a Student's-t negative log-likelihood can be justified by the generation of a bound on the update norm.

## A.2  PROOFS FOR SECTION 4.2

In this section we prove propositions (4.1)-(4.2).

First we show that finding the parameter that minimizes $\mathrm{KL}_t(\theta)$ is equivalent with finding the one that maximizes the conditional expectation of $p(y|\theta)$ with respect to the true statistical model or in other words we justify the second equality in Eq.(4):

$$
\begin{aligned}
\theta_t^* &= \underset{\theta \in \Theta}{\arg\min} \left[ \int_{\mathbb{R}^d} q_t(y) \log q_t(y) dy - \int_{\mathbb{R}^d} q_t(y) \log p(y|\theta) dy \right] \\
&= \underset{\theta \in \Theta}{\arg\min} \left[ - \int_{\mathbb{R}^d} q_t(y) \log p(y|\theta) dy \right] \\
&= \underset{\theta \in \Theta}{\arg\max} \, \mathbb{E}_{y \sim q_t} [\log p(y|\theta)]
\end{aligned}
\tag{9}
$$

*Proof of proposition (4.1).* From assumption (A1) we select $w_t^* \in W$ such that $f_{w_t^*}(x_t) = \theta_t^*$, then from Eq.(9) we observe that $\theta_t^*$ maximizes the expected log-likelihood with respect to $\theta$ under

$q_t$. Since $q_t$ is the true statistical model of the target time-series it corresponds to its empirical distribution, thus $\theta_t^*$ maximizes the plain log-likelihood,

$$\left.\frac{\partial \log p(y_t|\theta)}{\partial \theta}\right|_{\theta=\theta_t^*} = 0$$

and as a result $\nabla g_t(w_t^*) = 0$.

From $(A3)$, for $w_t, w_t^* \in W$ we get

$$\langle \mathscr{I}_t(w_t)^{-1}\nabla g_t(w_t) - \mathscr{I}_t(w_t^*)^{-1}\nabla g_t(w_t^*), w_t - w_t^* \rangle \leq -\frac{1}{c}\|\mathscr{I}_t(w_t)^{-1}\nabla g_t(w_t) - \mathscr{I}_t(w_t^*)^{-1}\nabla g_t(w_t^*)\|^2$$

$$\langle \mathscr{I}_t(w_t)^{-1}\nabla g_t(w_t), w_t - w_t^* \rangle \leq -\frac{1}{c}\|\mathscr{I}_t(w_t)^{-1}\nabla g_t(w_t)\|^2$$

$$\|\mathbb{E}_{t-1}[w_{t+1}] - w_t^*\|^2 = \|w_t + \eta\,\mathscr{I}_t(w_t)^{-1}\nabla g_t(w_t) - w_t^*\|^2$$
$$= \|w_t - w_t^*\|^2 + 2\langle \eta\,\mathscr{I}_t(w_t)^{-1}\nabla g_t(w_t), w_t - w_t^* \rangle + \eta^2\|\mathscr{I}_t(w_t)^{-1}\nabla g_t(w_t)\|^2$$
$$\leq \|w_t - w_t^*\|^2 - 2\frac{\eta}{c}\|\mathscr{I}_t(w_t)^{-1}\nabla g_t(w_t)\|^2 + \eta^2\|\mathscr{I}_t(w_t)^{-1}\nabla g_t(w_t)\|^2$$
$$= \|w_t - w_t^*\|^2 - \eta\left(\frac{2}{c} - \eta\right)\|\mathscr{I}_t(w_t)^{-1}\nabla g_t(w_t)\|^2.$$

We note that

$$\eta\left(\frac{2}{c} - \eta\right)\|\mathscr{I}_t(w_t)^{-1}\nabla g_t(w_t)\|^2 > 0$$

hence

$$\|\mathbb{E}_{t-1}[w_{t+1}] - w_t^*\| < \|w_t - w_t^*\|.$$

$\square$

*Proof of proposition (4.2).* From assumption (A1) we select $w_t^* \in W$ such that $f_{w^*}(x_t) = \theta_t^*$. Since we are interested in the properties of $f$ with respect to the weights we will slightly abuse the notation and write $f(w)$ instead of $f_w(x_t)$ for $w \in W$. From assumption (A4) there are constants $L, l > 0$ such that for any $w_1, w_2 \in W$

$$l\|w_1 - w_2\| \leq \|f(w_1) - f(w_2)\| \leq L\|w_1 - w_2\| \tag{10}$$

then we write

$$\|\theta_t(\mathbb{E}_{t-1}[w_{t+1}]) - \theta_t^*\| = \|f(\mathbb{E}_{t-1}[w_{t+1}]) - f(w_t^*)\|$$
$$\leq L\|\mathbb{E}_{t-1}[w_{t+1}] - w_t^*\|$$
$$< L\|w_t - w_t^*\|$$
$$\leq \frac{L}{l}\|f(w_t) - f(w_t^*)\| = \frac{L}{l}\|\theta_t(w_t) - \theta_t^*\|$$

the second inequality is due to the optimality of the weights (see proposition (4.1)). $\square$

## A.3 PROOF FOR SECTION 4.3

In order to lighten the notation we use the following conventions:

First, we omit the time index and the weight subscript thus we write $f(x)$ instead of $f_{w_t}(x_t)$.

The score that corresponds to the neural network (see section 4.1) is

$$\nabla_w(y) = \underbrace{\frac{\partial \log p(y|\theta)}{\partial \theta}}_{\nabla_\theta \log p(y|\theta)} \underbrace{\frac{\partial \theta}{\partial w}}_{J_w}$$

$$= J_w^\intercal \nabla_\theta \log p(y \mid \theta)$$

where $\frac{\partial \theta}{\partial w}$ is the Jacobian matrix and we denote it as $J_w$. Notice that every time we consider the score is before the weight update hence the gradient is with respect to the provisional output $f_{w_t}(x_t)$ and not the final (after the update) $f_{w_{t+1}}(x_t)$.

*Proof of Theorem 4.1.* The score, using the Tikhonov regularization (Martens, 2020) and the definition of the FIM, i.e. that is defined as the variance of the score, conditional on the input (Kunstner et al., 2019), it is:

$$\tilde{\nabla}_w \log p(y|f_w(x)) = \left(\mathbb{V}\big[J_w^T \nabla_{f(x)} \log p(y|f(x)) \mid x\big] + \tau\mathbf{I}\right)^{-1} \nabla_w(y)$$

$$= \left(\mathbb{V}\big[J_w^T \nabla_{f(x)} \log p(y|f(x)) \mid x\big] + \tau\mathbf{I}\right)^{-1} J_w^T \nabla_{f(x)} \log p(y|f(x)))$$

$$= \left(J_w^T \mathbb{V}\big[\nabla_{f(x)} \log p(y|f(x))) \mid x\big] J_w + \tau\mathbf{I}\right)^{-1} J_w^T \nabla_{f(x)} \log p(y|f(x)))$$

$$= \left(J_w^T \kappa\mathbf{I}J_w + \tau\mathbf{I}\right)^{-1} J_w^T \nabla_{f(x)} \log p(y|f(x))), \quad \kappa = \frac{v+1}{(v+3)s^2}$$

$$= \underbrace{V(\kappa\Sigma^T\Sigma + \tau\mathbf{I})^{-1}\Sigma^T U^T}_{B_1} \underbrace{\nabla_{f(x)} \log p(y|f_w(x)))}_{B_2}.$$

The third equality is due to the fact that the Jacobian matrix is conditionally independent from the input given the output.

The fourth equality is due to assumption of the Student's-t distribution

$$\mathbb{V}\big[\nabla_{f(x)} \log p(y|f(x))|x\big] = \frac{v+1}{(v+3)s^2}\mathbf{I}.$$

For the fifth equality we apply the SVD to the Jacobian matrix, i.e. $J_w = U\Sigma V^\mathsf{T}$, for $\Sigma$ diagonal, then taking the $L_2$-norm we get

$$\|\tilde{\nabla}_w \log p(y|f(x))\|_2 \leq \frac{1}{2\sqrt{\kappa\tau}} \|\nabla_{f(x)} \log p(y|f(x)))\|_2$$

$$\leq \frac{(v+1)\sqrt{m}}{4s\sqrt{\kappa\tau v}}$$

$$= \frac{1}{4}\sqrt{\frac{(v+1)(v+3)m}{\tau v}}$$

For the first inequality we compute the bound by using the definition of spectral norm as the maximum singular value of the matrix. The maximum is reached for $\sigma_i = \sqrt{\tau/\kappa}$.

$$\|B_1\|_2 = \|V(\kappa\Sigma^T\Sigma + \tau\mathbf{I})^{-1}\Sigma^T U^T\|_2 = \max_i \frac{\sigma_i}{\kappa\sigma_i^2 + \tau} \leq \frac{1}{2\sqrt{\kappa\tau}}.$$

The score of the Student's-t related to the output is:

$$B_2 = \nabla_{f(x)} \log p(y|f(x))) = -\left[\frac{(v+1)e_1}{vs^2 + e_1^2}, ...., \frac{(v+1)e_m}{vs^2 + e_m^2}\right],$$

with $m$ the number of outputs, $e_i = y_i - f(x)_i$ the error related to output $i$ and $v$ the degrees of freedom. □

## B  MODIFIED TIKHONOV REGULARIZER

With a regime change, the observed variance of the target conditional on the predictions will increase. We then want to also increase the assumed variance through the scale parameter $s$. The increase of $s$ needs to have an effect on the final update similar to what happens in standard score-driven models (see Figure 2): for $s \to \infty$ the update should converge to a linear function of the error as for the Gaussian assumption. To do this, we first write the natural gradient for the Student's t likelihood. Define $e = y - f_w(x)$

$$\tilde{\nabla}_w f_w(x) = \left(\frac{v+1}{(v+3)s^2} J_w^T J_w + \tau\mathbf{I}\right)^{-1} J_w^T \frac{(v+1)e}{vs^2 + e^2} =$$

$$= \left(\frac{v+1}{(v+3)} J_w^T J_w + s^2\tau\mathbf{I}\right)^{-1} J_w^T \frac{(v+1)e}{v + e^2/s^2}.$$

Hence, for the natural gradient update with the Tikhonov regularization, the limit for an infinite scale would be:

$$\lim_{s \to \infty} \tilde{\nabla}_w f_w(x) = 0,$$

which is clearly different from the linear function of $e$ we are aiming for. Hence, increasing the scale $s$ would not have the desired effect of monotonically accelerating learning. This is also confirmed by the fact that the bound in Eq. 5 does not depend on $s$. The culprit of this difference between the standard score-driven (Figure 2) and the natural gradient is the presence of the Tikhonov regularization. To recover the desired effect, propose to set $\tau = \frac{0.9\beta}{1+s^2} + \frac{0.1\beta}{s^2}$ with $\beta$ a scalar hyperparmeter. Note that the effective regularization added to the diagonal of the matrix $J_w^T J_w$ is $s^2 \tau \mathbf{I}$. With our particular choice of $\tau$, we obtain an effective regularizer $s^2 \tau \mathbf{I} = \frac{0.9\beta}{1/s^2+1} + 0.1\beta$ that is bounded in the interval $[0.1\beta, \beta]$, avoiding numerical instabilities when the scale is very small, but also avoiding regularizations that are too strong. After multiple experiments, we found this heuristic to be the most effective and safe. The limit with the new $\tau$ is:

$$\lim_{s \to \infty} \left( \frac{\nu+1}{(\nu+3)} J_w^T J_w + \left( \frac{0.9\beta}{1/s^2+1} + 0.1\beta \right) \mathbf{I} \right)^{-1} J_w^T \frac{(\nu+1)e}{\nu + e^2/s^2} =$$

$$\left( \frac{\nu+1}{(\nu+3)} J_w^T J_w + \beta \mathbf{I} \right)^{-1} J_w^T \frac{(\nu+1)e}{\nu},$$

obtaining a natural gradient that linearly grows with the error $e$ (as in the Gaussian case) when $s \to \infty$. The scale is now influencing the bound 5 through its effect on $\tau$: for a larger scale, we have larger update bounds, enabling fast adaptation.

## C    NATSR PRACTICAL IMPLEMENTATION

In our implementation of the method, we use some approximations and heuristics to make the process more efficient.

The FIM is approximated using Kronecker-Factored Approximate Curvature (K-FAC) (Martens & Grosse, 2015). This approximation greatly reduces the memory and computational requirements for inverting the FIM when computing the natural gradient. The gradient correlation between layers is ignored, and for each layer, only two small matrices are maintained and inverted: one for the outer product of the layer inputs and one for the outer product of the layer pre-activation gradients. These two Kronecker factors are estimated through an exponential moving average with a default smoothing factor of 0.5. This fast adaptation allows us to keep only local geometrical information.

The FIM is the expected value of the outer product of the gradient evaluated with respect to the output distribution, not the observed one (Kunstner et al., 2019). Following the approach of *nngeometry* (George, 2021), we estimate it through a Monte Carlo approach, taking $k$ samples from the predicted distribution, and evaluating the gradient of each. In this way, the computation of the FIM is independent of the output shape and can scale to larger output vectors.

To minimize the number of times the FIM needs to be computed and inverted, we reevaluate it only when necessary. When not updated, it simply corresponds to the one used at the previous step. Our heuristics trigger the update when the currently observed loss is in the worst $p\%$ of recently observed losses or, anyway, after a fixed number of steps to avoid situations where the FIM is never updated. The distribution of recently observed losses is estimated assuming a Normal distribution and keeping track of two additional exponential moving averages for the mean and the variance of the loss history.

## D    HYPERARAMETER SENSITIVITY

To clarify the influence of NatSR's hyperparameters, we conducted a sensitivity analysis where we vary four key components: (i) the EMA smoothing parameter $\alpha$, (ii) the Tikhonov regularization strength $\tau$, (iii) the replay buffer size, and (iv) the degrees of freedom $\nu$ of the Student's t distribution. All variations are measured relative to the hyper-optimized configuration, which serves as our baseline. Table 3 reports the resulting changes in performance.

| | $\alpha$ | | | | | $\tau$ | | | | | buffer size | | | | | $\nu$ | | | | |
|---|---|---|---|---|---|---|---|---|---|---|---|---|---|---|---|---|---|---|---|---|
| | 0.1 | 0.3 | 0.55* | 0.7 | 0.99 | 0.01 | 0.14* | 0.5 | 0.9 | 1.5 | 2 | 4 | 8* | 16 | 32 | 10 | 25 | 50* | 250 | 500 |
| **ETTh1** | -.002 | -.009 | 0.0 | -.008 | -.002 | $9.7\cdot10^{8}$ | 0.0 | .023 | .053 | .098 | .020 | .011 | 0.0 | -.013 | -.009 | -.008 | -.010 | 0.0 | -.008 | -.002 |
| **ETTm1** | .004 | -.001 | 0.0 | .002 | .013 | $+\infty$ | 0.0 | .030 | .079 | .154 | .014 | .017 | 0.0 | .000 | -.001 | .011 | .006 | 0.0 | -.001 | -.003 |
| **WTH** | .005 | .004 | 0.0 | .009 | .038 | $+\infty$ | 0.0 | .019 | .053 | .093 | .063 | .026 | 0.0 | .003 | .005 | -.003 | .005 | 0.0 | .017 | .030 |

Table 3: Performance sensitivity analysis for each hyperparameter. Each entry is the $\Delta$MASE between a configuration and the hyper-optimized one (marked with *). We used infinity when training did not converge.

Among all the parameters, the Tykhonov regularization $\tau$ has by far the strongest impact. When $\tau$ is too high, the regularization overwhelms the adaptation capabilities of NatSR. On the contrary, setting $\tau$ too small destabilizes the training process and can even cause it to fail. Similar sensitivity has been widely noted in natural gradient and other curvature-based learning methods (Martens, 2020). In our setting, we noticed that finding the sweet spot can is both dataset-dependent and particularly challenging for rapidly evolving time series.

The remaining hyperparameters show smoother and more predictable behavior. Variations in $\alpha$ and $\nu$ produce only minor deviations across datasets, always consistent with their role in balancing stability and adaptation. The replay buffer size, in contrast, exhibits unique behavior. Typically, especially for stationary datasets, larger buffers improve performance, as using more data to compute the updates stabilizes the gradients and likelihood estimation. However, overly large replay buffers can limit responsiveness to distributional shifts, degrading performance on highly non-stationary datasets. This limitation can be mitigated with more advanced replay strategies, but this trade-off remains an important consideration.

# E    NatSR algorithm

---

**Algorithm 1:** Natural Score-driven Replay (NatSR)

---

**Input:** network parameters $w$; learning rate $\eta$; EMA parameter $\alpha_{\text{EMA}}$; memory importance $\lambda$;
      degrees of freedom $\nu$; regularizer $\beta$; scale learning rate $\eta_s$.

$\mathscr{B} \leftarrow \varnothing$

$s \leftarrow 1$

$\mathscr{L}_w(D_t, s^2; \nu) = \frac{1}{|D|} \sum_{\{X_i, y_i\} \in D_t} \frac{\nu+1}{2} log\left(1 + \frac{(y_i - f_w(X_i)^2)}{\nu s^2}\right)$        `// Student's t loss function`

**for** $t \leftarrow 1, 2, \ldots$ **do**

    $N_t = \{X_t, y_t\}$          `// Obtaining the new observation`

    $B_t \subseteq \mathscr{B}$          `// Sample batch from Buffer of past data`

    $L \leftarrow \mathscr{L}_w(N_t, s; \nu) + \lambda \mathscr{L}_w(B_t, s; \nu)$      `// Compute the loss on New and Buffer data`

    $\nabla_w L$          `// Compute gradient`

    $\tau \leftarrow \frac{1}{\beta + s^2}$          `// Define Tikhonov regularizer`

         `// τ is a function of the scale s`

    **if** *L worst* 1% *of recent Ls* **then**

        | Update FIM $\leftarrow$ *True*

    **else**

        | Update FIM $\leftarrow$ *False*

    **end**

    **if** *Update FIM* **then**

        Compute Monte Carlo K-FAC factors $A$ and $G$ from $N_t$ and $B_t$ (weight $B_t$ by $\lambda$)

        **if** $F_{EMA} \neq \varnothing$ **then**

            **for** $l \leftarrow 1$ **to** $L$ **do**

                $A_{\text{EMA},l} \leftarrow (1 - \alpha_{\text{EMA}}) A_{\text{EMA},l} + \alpha_{\text{EMA}} A_l$      `// EMA for the KFAC factors`

                $G_{\text{EMA},l} \leftarrow (1 - \alpha_{\text{EMA}}) G_{\text{EMA},l} + \alpha_{\text{EMA}} G_l$      `// EMA for the KFAC factors`

            **end**

            $F_{\text{EMA}} \leftarrow \{A_{\text{EMA}}, G_{\text{EMA}}\}$

        **else**

            $F_{\text{EMA}} \leftarrow \{A, G\}$          `// If no EMA, initialize it`

        **end**

        $F_{\text{INV}} \leftarrow \left(F_{\text{EMA}} + \tau \mathbf{I}\right)^{-1}$          `// Inverse FIM with Tikhonov regularization`

    **end**

    $\tilde{\nabla}_w L \leftarrow F_{\text{INV}} \nabla_w L$          `// Natural Gradient Update`

    $s^2 \leftarrow s^2 + \eta_s \frac{1}{|N_t| + |B_t|} \sum_{\{X_i, y_i\} \in N_t, B_t} \frac{\nu s^2 [(y_i - f_w(X_i))^2 - s^2]}{\nu s^2 + (y_i - f_w(X_i))^2}$    `// Score driven update of the scale`

    **if** *optimizer is Adam* **then**

        | $\tilde{\nabla}_w L \leftarrow \text{AdamUpdate}\tilde{\nabla}_w L$

    **end**

         `// Use ADAM optimizer`

    $w \leftarrow w - \eta \, \tilde{\nabla}_w L$

    $\mathscr{B} \leftarrow \text{ReservoirUpdate}(\mathscr{B}, N_t, \text{maxsize})$          `// Update the Buffer`

**end**

---

# F    Additional Experimental Details

During the online phase, the batch size is set to 1, so the model is trained and evaluated at each new observation. In addition to this, methods using memory buffers sample 8 samples from the buffer.

All methods undergo a hyperparameter optimization repeated 30 times on a complete online learning with the ETTh1 dataset. During this phase, we select the best online learning rate, and keep it fixed when the methods are tested on the other datasets. For NatSR, also the best $\alpha_{EMA}$ used for the estimation of the gradient and the FIM is selected. The values of method-specific hyperparameters are the same as the ones reported in the original papers and the available code of Pham et al. (2023) and Wen et al. (2023).

The number of features to predict depends on the dataset and can go from as few as 7 for ETT datasets to as many as 862 for Traffic. The length of the input time series is always set to 60, while the prediction length can be 1, 24, or 48. The only exception to this is Traffic, for which we excluded the value 48 as in Pham et al. (2023), due to the huge number of features of the dataset.

In terms of hardware, all experiments are executed on a Linux machine equipped with two Tesla V100 16GB GPUs and Intel Xeon Gold 6140M CPUs.

**Backbone architecture:** All strategies use a Temporal Convolutional Network (TCN) (Bai et al., 2018) as a backbone. The sizes of the networks are the same, except for FSNET, which modifies the architecture with internal layers for the learning of adaptation coefficients, and OneNet, which keeps two separate TCNs, one doing convolutions only on the temporal dimension and one only on the variables' dimension. We test both Mean Squared Error and Mean Absolute Error losses.

**Evaluation Metric:** Choosing the correct metric to compare the different methods is not an easy task. In this paper we follow the choice of Monash repository (Godahewa et al., 2021), probably the most extensive open-source comparison of forecasting models, of using the Mean Absolute Scaled Error (MASE) to compare methods (Hyndman & Koehler, 2006). It is defined as the mean absolute error of the forecasting model, divided by the mean absolute error of the one-step naive forecaster. MASE is symmetric for positive and negative errors, scale-invariant, robust to outliers, and interpretable. For these reasons, it is considered a solid choice to compare different approaches (Franses, 2016). Note that values greater than 1 do not imply the naive forecaster is better, since it makes only one-step-ahead predictions while the models forecast multiple steps ahead.

**NatSR experimental setup:** When testing NatSR, we evaluate the FIM by sampling $k = 100$ samples from the predicted distribution. This evaluation is performed only when the observed loss is in the worst 1% of recently observed losses, estimating the mean and the variance of recently observed losses by using two EMAs with 0.01 weight for new observations. The dynamic scale is updated by a score-driven model using a learning rate $\alpha_s = 0.1$. The $\eta$ is fixed to 1, hence bounding the maximum Tikhonov regularizer $\tau$ to 1. During the online phase, we tried two different approaches for the optimizer: $v = 50$ and SGD, and $v = 500$ and AdamW. The first combination (NatSR$_{stable}$) gives a more robust method with stricter bounds on the norm of the updates, while the second (NatSR$_{fast}$) allows for bigger updates, but it also introduces Adam's empirical normalization to stabilize the process. The Buffer Size is of 500 samples, the same used in ER.

# G   ADDITIONAL EXPERIMENTAL RESULTS

| Dataset | Pred. Len | OGD | ER | DER++ | FSNET$_{MAE}$ | OneNet$_{MAE}$ | NatSR (Ours) |
|---------|-----------|-----|-----|-------|--------|---------|--------------|
| ECL   | 1  | 3.02 | 2.55 | 3.35 | 0.41 | 0.12 | 2.15 |
|       | 24 | 2.52 | 1.96 | 0.86 | 1.29 | 0.04 | 0.60 |
|       | 48 | 2.89 | 2.20 | 0.73 | 0.36 | 0.14 | 0.11 |
| ETTh1 | 1  | 0.50 | 0.62 | 0.65 | 0.66 | 0.15 | 0.36 |
|       | 24 | 1.31 | 1.44 | 1.30 | 2.16 | 0.35 | 0.05 |
|       | 48 | 0.75 | 1.00 | 0.95 | 1.01 | 0.88 | 1.00 |
| ETTh2 | 1  | 0.43 | 0.60 | 0.60 | 0.89 | 0.18 | 0.40 |
|       | 24 | 1.44 | 0.41 | 0.53 | 1.24 | 0.98 | 0.89 |
|       | 48 | 1.05 | 1.21 | 0.95 | 2.05 | 0.32 | 2.95 |
| ETTm1 | 1  | 0.81 | 0.44 | 0.45 | 1.94 | 0.38 | 0.24 |
|       | 24 | 1.09 | 1.34 | 4.13 | 3.32 | 1.98 | 1.62 |
|       | 48 | 2.25 | 3.80 | 7.08 | 2.41 | 0.66 | 2.87 |
| ETTm2 | 1  | 1.04 | 0.51 | 0.43 | 0.98 | 0.24 | 0.25 |
|       | 24 | 1.23 | 0.41 | 0.86 | 0.43 | 2.04 | 0.44 |
|       | 48 | 2.33 | 1.73 | 2.37 | 1.90 | 0.65 | 0.34 |
| Traffic | 1 | .... | .... | .... | 0.75 | 0.11 | 0.50 |
|         | 24 | 0.34 | 0.13 | 0.19 | 0.59 | 0.16 | 0.70 |
| WTH   | 1  | 0.82 | 0.41 | 0.77 | 0.42 | 0.14 | 0.15 |
|       | 24 | 1.05 | 0.21 | 0.62 | 1.10 | 0.58 | 0.68 |
|       | 48 | 1.19 | 0.86 | 0.98 | 0.58 | 0.40 | 1.18 |

Table 4: Standard Deviations of MASE multiplied by 100 for MSE loss.

| Dataset | OGD | ER | DER++ | FSNET$_{MAE}$ | OneNet$_{MAE}$ | NatSR (Ours) |
|---------|-----|-----|-------|--------|---------|--------------|
| ECL   | 139 | 241 | 233 | 1111 | 816  | 1185 |
| ETTh1 | 71  | 124 | 127 | 627  | 457  | 588  |
| ETTh2 | 70  | 122 | 122 | 601  | 434  | 610  |
| ETTm1 | 73  | 128 | 122 | 637  | 428  | 575  |
| ETTm2 | 116 | 231 | 222 | 663  | 417  | 474  |
| Traffic | 141 | 231 | 210 | 804  | 600  | 1909 |
| WTH   | 177 | 298 | 286 | 1541 | 1043 | 1268 |

Table 5: Mean total online training time in seconds for prediction length 24.

| Dataset | Pred. Len | OGD | ER | DER++ | FSNET | FSNET$_{MAE}$ | OneNet | OneNet$_{MAE}$ | NatSR$_{stable}$ | NatSR$_{fast}$ |
|---|---|---|---|---|---|---|---|---|---|---|
| ECL | 1 | 1.67 | 1.43 | 1.11 | 1.56 | 0.96 | 0.78 | **0.64** | 3.53 | 0.78 |
| | 24 | 3.12 | 3.21 | 2.89 | 2.99 | 1.42 | 1.14 | **0.92** | 4.01 | 1.41 |
| | 48 | 3.27 | 3.01 | 2.96 | 3.12 | 1.44 | 1.17 | **0.96** | 4.14 | 1.55 |
| ETTh1 | 1 | 0.87 | 0.83 | 0.82 | 0.93 | 0.92 | 0.86 | 0.83 | **0.79** | 0.83 |
| | 24 | 1.50 | 1.49 | 1.45 | 1.05 | 1.08 | 1.42 | 1.38 | **0.97** | 1.40 |
| | 48 | 1.46 | 1.42 | 1.42 | 1.15 | 1.16 | 1.38 | 1.39 | **1.12** | 1.37 |
| ETTh2 | 1 | 1.14 | 1.09 | 1.08 | 1.11 | 1.10 | 1.12 | 1.06 | **1.01** | 1.08 |
| | 24 | 1.65 | 1.60 | 1.60 | 1.36 | 1.37 | 1.52 | 1.56 | **1.23** | 1.64 |
| | 48 | 1.63 | 1.62 | 1.61 | 1.49 | 1.48 | 1.54 | 1.61 | **1.39** | 1.64 |
| ETTm1 | 1 | 1.18 | 1.03 | 1.01 | 1.06 | 1.10 | 1.11 | 1.05 | **0.97** | 1.03 |
| | 24 | 2.19 | 1.82 | 1.83 | 1.39 | 1.45 | 1.86 | 1.91 | **1.16** | 1.83 |
| | 48 | 2.19 | 1.91 | 1.81 | 1.51 | 1.52 | 1.95 | 2.04 | **1.33** | 1.83 |
| ETTm2 | 1 | 1.36 | 1.26 | 1.24 | 1.27 | 1.14 | 1.33 | 1.15 | **1.12** | 1.24 |
| | 24 | 2.03 | 1.80 | 1.81 | 1.46 | 1.48 | 1.80 | 1.79 | **1.37** | 1.82 |
| | 48 | 2.01 | 1.85 | 1.82 | 1.52 | 1.51 | 1.87 | 1.84 | **1.50** | 1.80 |
| Traffic | 1 | 0.84 | 0.79 | 0.78 | 0.78 | 0.70 | 0.68 | **0.62** | 0.89 | 0.71 |
| | 24 | 1.05 | 1.07 | 0.95 | 0.95 | 0.96 | 0.97 | 0.91 | 1.14 | **0.87** |
| WTH | 1 | 1.47 | 1.30 | 1.44 | 1.17 | 1.19 | 1.15 | 1.06 | **1.04** | 1.25 |
| | 24 | 1.98 | 1.90 | 1.84 | 1.39 | 1.44 | 1.80 | 1.73 | **1.25** | 1.80 |
| | 48 | 1.97 | 1.89 | 1.86 | 1.45 | 1.46 | 1.85 | 1.82 | **1.39** | 1.86 |

Table 6: Complete table of average MASE across 3 runs. Best in **bold**, second best underlined.