# OpenReview forum: "Online Continual Learning for Time Series: a Natural Score-driven Approach"
_ICLR.cc/2026/Conference — Submitted to ICLR 2026_

### Official Review · Reviewer_Nj1d · 2025-10-31

**Soundness:** 3
**Presentation:** 3
**Contribution:** 2
**Rating:** 4
**Confidence:** 3

**Summary:**

This paper builds the connection between natural gradient descent and the score-driven model, or the generalized autoregressive score (GAS) model. It shows that natural gradient descent is a score-driven model and proves its information-theoretica optimality. To improve robustness to outliers, it introduces the student’s likelihood and shows the bounded update. It is further combined with a replay buffer and a dynamic scale heuristic called natural score-driven replay (NatSR). Overall, limited experiments are performed to test the proposed algorithm.

**Strengths:**

1.	This paper is a theoretical paper. The biggest contribution of this paper is that it reveals the connection between natural gradient descent and the score-driven model, or the generalized autoregressive score (GAS) model. From this connection, the student’s likelihood and a replay buffer are introduced, leading to a a dynamic scale heuristic called natural score-driven replay (NatSR). From theoretical perspective, this paper is a good addition to the time series community.

**Weaknesses:**

1.	Except the theoretical results, the paper looks weak, especially its practical value in real-world applications. For example, the experiments are quite limited, with limited data, limited baselines and only one metric. Only MASE is reported. It is suggested to report RMSE and MAPE as well. In particular, on some datasets (ECL and Traffic), the performance of the proposed method is much worse than other baselines. I am not sure if the current experiments can justify the publication of this paper.

**Questions:**

1. Can you expand the experiments, including more SOTA baselines, more evaluation metrics, and more datasets. Given the current experimental results, it is fairly hard to me to justify the publication of this paper.

---

> ### Author Response · Authors · 2025-11-22
>
> Dear Reviewer, thank you for the kind words about our theoretical contributions. Indeed, our primary goal was to propose a theoretically grounded approach inspired by classical time-series methodologies, something that we think is lacking in the deep learning forecasting community. For this reason, we do not feel that increasing the number of experiments would substantially strengthen the contribution of our work, which is not empirical in nature. Experiments are used only to verify the strengths and weaknesses of the theoretical idea. We would also like to highlight the following points:
> - Breadth of evaluation: We evaluated our method on more real-world datasets than those used in the SOTA baselines we compare against (FSNet, OneNet), both of which were accepted at top-tier conferences.
> - Architectural constraints of baselines: FSNet and OneNet are strictly tied to the TCN backbone. Evaluating our method on different backbones would make direct comparison less meaningful, as the baselines cannot be adapted without altering their core design.
> - Metrics: We justified the use of MASE as a metric in Appendix F. We also want to underline that recent and important works are following the same approach of evaluating the point forecast only through MASE [1,2].
>
> Finally, we believe that showing also the datasets where our method is not performing well is actually a strength of our paper: it demonstrates transparency and helps identify the true limitations of our approach for future research.
>
> [1] Ansari, Abdul Fatir et al. “Chronos-2: From Univariate to Universal Forecasting.” ArXiv abs/2510.15821 (2025) \
> [2] Liu, Chenghao et al. “Moirai 2.0: When Less Is More for Time Series Forecasting.” (2025).

---

### Official Review · Reviewer_xQJz · 2025-11-01

**Soundness:** 3
**Presentation:** 2
**Contribution:** 3
**Rating:** 4
**Confidence:** 2

**Summary:**

This paper proposes Natural Score-driven Replay (NatSR), a novel method for Online Continual Learning (OCL) applied to Time Series Forecasting (OTSF). The core idea is to bridge concepts from econometrics (score-driven models) and optimization (natural gradient descent) to create a more robust and theoretically grounded online learner. It also integrates this with an experience replay buffer, resulting in a practical OCL algorithm that achieves competitive performance on several real-world time series datasets.

**Strengths:**

**Originality:** The conceptual link between natural gradient descent and score-driven models is a significant and novel insight. It provides a fresh, unifying perspective on online optimization.

**Clarity:** Despite the technical density in some sections, the core ideas are effectively communicated. The abstract and introduction succinctly outline the contributions, and the experiments are clearly described.

**Significance:** The paper tackles a fundamental problem (online learning in non-stationary environments) with a method that is both principled and practical. Its strong performance on real-world data, without requiring specialized architectures, makes it a valuable contribution for the broader ICLR community interested in robust and adaptive machine learning.

**Weaknesses:**

**Presentation:** Some parts, particularly in Sections 4.2 and 4.3, are quite dense and could be challenging for a reader not deeply familiar with both information geometry and score-driven models. A little more high-level intuition before diving into the equations would improve accessibility. The pseudocode in Appendix D is helpful but could be more clearly annotated.

**Performance on High-Dynamics Datasets:** The method underperforms on the ECL and Traffic datasets, which the authors attribute to a need for more plasticity. While this is a fair observation, it highlights a limitation: the conservatism induced by the robustness guarantee can be a drawback in certain environments. The paper would be strengthened by a deeper analysis or a more adaptive mechanism for this trade-off, perhaps beyond the NatSR_fast variant mentioned in the appendix.

**Theoretical Assumptions:** The theoretical optimality results (Propositions 4.1 and 4.2) rely on assumptions like the Bi-Lipschitz condition (A4). A discussion on how realistic these assumptions are for deep neural networks in practice would add nuance to the theoretical contributions.

**Questions:**

- The heuristic for updating the FIM (only on the worst 1% of losses) is pragmatic but somewhat divorced from the core theory. How sensitive are the results to this heuristic (e.g., the 1% threshold)? Was there any exploration of more theoretically motivated triggers, such as those based on the norm of the score or detected regime shifts?

- The bounded update property is reminiscent of proximal methods or gradient clipping. Could the authors provide more discussion or a small experiment comparing the type of robustness achieved by NatSR versus a carefully tuned proximal method or a simple combination of AdamW with gradient clipping and a Student's t-loss?

---

> ### Author Response · Authors · 2025-11-22
>
> Dear Reviewer, thank you very much for your effort and comments. Below, we provide our detailed responses to the points you raised.
>
> Presentation issues:
>
> Thank you for raising this point. We agree with your comments, so we added some bullet points to summarize the findings of the paper to give a broader perspective in the introduction. We also rewrote some parts of sections 4.1 and 4.2, adding some higher-level intuition. We modified the pseudocode in Appendix D, adding more annotations.
>
> Issues about the Performance on High-Dynamics Datasets:
>
> We agree about the existence of this trade-off, which in continual learning terminology, is often refered as the stability-plasticity dilemma. Our method naturally leans toward stability in its standard form, but we showed that it is possible to tune it for faster adaptation if necessary. If the reviewer thinks it would improve the paper, we can for sure move the explanation and results of the NAtSR$_{\text{fast}}$ to the main paper, adding a deeper explanation. Basically, the idea is that by adjusting the degrees of freedom, it is possible to control the bound proposed in Section 4.3, allowing to choose a more/less adaptive optimization. Unfortunately, we do not think it is possible to find a single bound that works for every dataset, as there will be some datasets full of outliers that need to be ignored, and others where unexpected observations are actually regime shifts. We tried multiple possible adaptive mechanisms, but we always found the same problem: sometimes it is better to ignore extreme shifts as they are outliers, while in other cases, extreme shifts are actually real shifts. We believe there is still no possibility of classifying them without seeing the future.
>
> Theoretical Assumptions:
>
> We updated our paper with a more specific comment on the bi-Lipschitz condition. In any case, we want to highlight that such an assumption is needed only for Proposition 4.2. Thanks for the suggestion.
>
> Answer to Question 1:
>
> We agree that such a heuristic is not theoretically grounded. The actual reason for it is only to speed up the process, without excessively harming the accuracy of the FIM. We tried multiple different possibilities, but in the end, we opted for the simplest one, as we believe it keeps the focus on the optimizer and is easily interpretable. The basic idea is that the FIM needs to be recomputed in two cases: when a regime shift happens, and when it becomes outdated, to help the optimization. We believe that our approach is working for both scenarios. First, we are actually estimating the mean and variance of the historical losses, and then using a statistical test to check if the current loss is significantly worse than the previous ones. In this sense, we believe this is actually a regime shift detector. Moreover, not only can it catch the shifts in the data, but it can also trigger a FIM update if, even on the same regime, it checks that the optimization is not working anymore. For these reasons, we believe it is a useful heuristic that does not add excessive overhead.
>
> Answer to Question 2:
>
> The fundamental characteristic of NatSR, linked to score-driven models, is found both in the use of the FIM as a gradient preconditioner and in the choice of the loss (Student's t). The Student's t can bound the gradient as a gradient clipping, but with a theoretically justifiable statistical assumption about the data. The more substantial difference, however, derives the use of the FIM. The FIM induces a Riemannian Geometry, where distances in the parameter space are not measured in the standard Euclidean metric, but using the KL divergence, which is a more appropriate way to measure distances between distributions. Hence, the gradient direction is fundamentally transformed to account for the curvature of this geometry.. This approach has been greatly justified thanks to the work of Amari on Information Geometry [1]. When combined with the use of the Tikhonov regularization (basically an $l_2$ penalty on the weights), our method can actually be interpreted as a proximal method in a non-Euclidean geometry. In contrast, gradient clipping in the standard Euclidean space would not actually change the direction of the gradient, but would only control its norm. With the FIM, we can penalize specific directions that are more important for past tasks (higher curvature), hence improving the whole optimization process.
>
> [1] Amari, Shun-ichi. Information geometry and its applications. Vol. 194. Springer, 2016.

---

### Official Review · Reviewer_ALYR · 2025-11-03

**Soundness:** 3
**Presentation:** 2
**Contribution:** 2
**Rating:** 4
**Confidence:** 2

**Summary:**

This paper studies online continual learning (OCL) for time-series prediction. First, it analyzes neural network optimization from a score-driven parameter filtering perspective. Next, it uses a Student-t likelihood to bound update norms for robustness. Finally, it proposes NatSR, which combines natural gradient descent, a replay buffer, and a dynamic scale heuristic.

**Strengths:**

1. The paper tackles an important and practical OCL setting in time-series prediction—requiring high plasticity to adapt while maintaining high stability to avoid forgetting.
2. The analysis of natural gradient descent from a score-driven filtering perspective is reasonable and original.
3.  The proposed method is simple and useful, combining natural gradients, replay, and a dynamic scale heuristic in a coherent recipe.

**Weaknesses:**

1. Baselines are limited to rehearsal-style OCL. There are additional families relevant to OCL in time series: Error-feedback methods (e.g., online model adaptation, Extended Kalman Filter) and feedforward adaptation approaches that pair memory buffers with sample selection and tailored optimization [1]. Regularization-based continual learning methods to mitigate catastrophic forgetting [2]. Including these baselines (or carefully justifying their omission) would strengthen the empirical
2. The neural backbone is no longer state-of-the-art or standard in current time-series forecasting. Since the backbone strongly affects performance, results on stronger models would be more convincing (e.g.,  [3,4]).
3. The proposed second-order updates can add overhead. It would be useful to report time/space costs for NatSR and baselines.
4. The rationale for key hyperparameters is unclear. Please provide ablation/sensitivity for $k, \tau, \nu$, and buffer size.
5. Minor. Recent work studies unified/foundation time series prediction models [5]. Please discuss when OCL is preferable compared to training a large time-series foundation model and deploying it frozen. What benefits does OCL offer in deployment?

[1] Abuduweili, Abulikemu, and Changliu Liu. "Online model adaptation with feedforward compensation." Conference on Robot Learning. PMLR, 2023.
[2] Zhao, Xuyang, et al. "A statistical theory of regularization-based continual learning." arXiv preprint arXiv:2406.06213 (2024).
[3] Zeng, Ailing, et al. "Are transformers effective for time series forecasting?." Proceedings of the AAAI conference on artificial intelligence. Vol. 37. No. 9. 2023.
[4] Lin, Shengsheng, et al. "Cyclenet: Enhancing time series forecasting through modeling periodic patterns." Advances in Neural Information Processing Systems 37 (2024): 106315-106345.
[5] Woo, Gerald, et al. "Unified training of universal time series forecasting transformers." (2024): 53140.

**Questions:**

1. Could you add feedback/feedforward and regularization-based baselines, or provide a clear rationale if not feasible?
2. Could you evaluate on a stronger backbone (e.g., Patch/Transformer-style, periodic models) to test NatSR’s generality?
3. What is the time and memory cost of NatSR vs. baselines during the online phase?
4. Please report hyperparameter sensitivity for  $k, \tau, \nu$, and buffer size.
5. How do you justify OCL relative to unified/foundation time-series models? When is OCL the better choice

---

> ### Author Response · Authors · 2025-11-22
>
> Dear Reviewer, thank you for your time and your feedback. Please find below our responses to your comments and requests.
>
> Answer 1:
>
> Our choice to use rehearsal methods comes from the large set of OCL literature that found replay methods to always outperform other families of approaches [1,2]. Pure regularization approaches, for example, frequently need knowledge about the task boundary to actually penalize the weights important for previous tasks. In our paper, we tested DER++, which actually also uses a regularization approach in conjunction with experience replay. Regarding error-feedback methods, we actually think that all our tested methods rely on an online error adaptation through gradient descent. NatSR in particular can be reframed as a filtering algorithm equivalent to a score-driven model. Score-driven models adjust their assumptions following the direction that minimizes the currently experienced errors (like in an error-feedback adaptation). While our paper showed the equivalence between natural gradient and score-driven models, another work by Ollivier [3] showed exactly the equivalence between natural gradient and Kalman filter, hence strengthening our main point. We added the paper "Online Model Adaptation with Feedforward Compensation" in the related works, as we believe it is highly relevant and can be explored in future works.
>
> [1] Bidaki, Seyed Amir et al. “Online Continual Learning: A Systematic Literature Review of Approaches, Challenges, and Benchmarks.” ArXiv abs/2501.04897 (2025) \
> [2] Soutif-Cormerais, Albin et al. “A Comprehensive Empirical Evaluation on Online Continual Learning.” 2023 IEEE/CVF International Conference on Computer Vision Workshops (ICCVW) (2023): 3510-3520.
> [3] Ollivier, Yann. “Online natural gradient as a Kalman filter.” Electronic Journal of Statistics 12 (2017): 2930-2961.
>
> Answer 2:
>
> Our paper is focused on proposing an optimization algorithm with a strong theoretical foundation from time series analysis, as we believe it is still missing from the forecasting community. As such, we believe that our work is independent of the specific architectural choice. At the same time, as OCL for time series is still a very young research direction, the only SOTA baselines (FSNet and OneNet) are strictly linked to the use of TCN backbone and cannot be adapted to other architectures without changing their core mechanism. Therefore, using a different backbone would make a fair comparison nearly impossible. Finally, even in standardized third-party benchmarks [4], there is still no clear evidence that any single architecture consistently outperforms others.
>
> [4] Monash Time Series Forecasting Repository https://forecastingdata.org/
>
> Answer 3:
>
> In Table 4 in Appendix G, we report the average training times for the online phase. Our method adds an overhead that is comparable to SOTA approaches like FSNET.
>
> Answer 4:
>
> We believe this is a very significant request. We added a performance sensitivity analysis section in the appendix (Section D), also showing the selected hyperparameters for our experiments. We added the buffer size information to the appendix (500 samples like ER). Thanks for the suggestion.
>
> Answer 5:
>
> Unlike vision and language, time series have a lot of diversity and noise, frequently needing specific domain knowledge for each dataset. While time series foundation models are a very promising direction, they do not remove the need for finetuning [4,5,6], which would need to be carefully performed to not forget the pretraining information, and, in case of online streams of data, the process would be quite similar to the OCL setting. In fact, we would argue that researching stable continual finetuning methods will be even more important once foundation models become more popular, as they will nicely complement the pretraining process. Our paper proposes a first step for a time-series theoretically grounded optimization process, which would be interesting to extend to a finetuning pipeline in future works.
>
> [4] Karaouli, Nouha, et al. "How Foundational are Foundation Models for Time Series Forecasting?." arXiv preprint arXiv:2510.00742 (2025). \
> [5] X. Fu, M. Hirano and K. Imajo, "Financial Fine-Tuning a Large Time Series Model," 2025 IEEE Symposium on Computational Intelligence for Financial Engineering and Economics (CiFer), Trondheim, Norway, 2025, pp. 1-9, doi: 10.1109/CiFer64978.2025.10975735. \
> [6] Faw, M., Sen, R., Zhou, Y. & Das, A.. (2025). In-Context Fine-Tuning for Time-Series Foundation Models. Proceedings of the 42nd International Conference on Machine Learning, https://proceedings.mlr.press/v267/faw25b.html.

---

### Official Review · Reviewer_Pftj · 2025-11-10

**Soundness:** 2
**Presentation:** 2
**Contribution:** 2
**Rating:** 6
**Confidence:** 2

**Summary:**

This paper proposes Natural Score-Driven Replay (NatSR), a new method for online continual learning (OCL) applied to time series forecasting (OTSF). The key insight is to reinterpret natural gradient descent as a score-driven filtering process, linking it theoretically to Generalized Autoregressive Score (GAS) models from econometrics. The authors then discuss the proposed method with theoretical justifications.  The authors also use empirical experiments to show the performance improvement over the existing methods.

**Strengths:**

1. The connection between natural gradient descent and score-driven models is interesting and seems new in time series forecasting literature.

2. Consistent gains over strong baselines on several datasets; the ablation study supports the claims.

3. Sample code is provided to help the reviewer verify the results independently.

**Weaknesses:**

1. The writing could be improved. For example, the main algorithm, which, from my perspective, is one of the most important parts of this work, is deferred to the appendix. Moreover, it would be better to clearly include a separate section or use bullet points to state the major contributions.

2. The numerical experiments are limited to end-to-end trained models. The current experimental settings largely follow Pham et al. (2023)
and Wen et al. (2023). The small models are end-to-end trained on each single dataset. Recently, various pretrained time-series foundation models (e.g., models in GIFT-EVAL benchmark) provided even stronger baselines than those end-to-end models. It would be better extend the current algorithm into fine-tuning pretrained time-series foundation models.

**Questions:**

Please see the weakness part.

At the current stage, I consider this paper borderline, though I lean toward recommending acceptance. I remain open to re-evaluating my assessment after further discussions.

**Details Of Ethics Concerns:**

N.A

---

> ### Author Response · Authors · 2025-11-22
>
> Dear Reviewer,
> Thank you so much for your review. We hope to be able to address your request:
>
> Answer 1:
>
> We agree with the reviewer that having the algorithm in the main paper would be better but unfortunately, it is quite long and we did not find a way to fit it into the main text without sacrificing other important parts. We tried to improve it by annotating it more. We added the bullet points for the main contributions of the paper in the introduction. Thanks for the suggestion.
>
> Answer 2:
>
> We greatly appreciate this idea. In fact, we are currently exploring finetuning of foundation models for a separate project. Unfortunately, this setting has some peculiarities:
>  - First, a foundation model is not just an architectures, but a full processing pipelines with normalization steps and custom output heads (like the quantile head of Chronos 2). This makes it difficult to disentangle the role of the optimizer from the rest of the components and hard to integrate custom losses without changing some of the fundamental ideas of such pipelines.
>  -  Our method is fundamentally an optimization algorithm with a theoretical backing from time series analysis. We strongly believe that the only way to see if an optimization algorithm works, is to test it on an end-to-end training, to make it clear if there are some basic problems at the beginning or final part of the training process. Finetuning a pre-trained model would only test such algorithms in the final part of the training, where only few parameters need to be adjusted to the specific task.
>  -  On our preliminary tests, we found that on specific datasets (like FRED-MD), a simple feedforward network can obtain similar results to finetuned foundation models if the data are correctly prepared, challenging the dominance of foundation models on specific datasets.
>  -  Unfortunately, online continual learning for time series is a very young and unexplored field. The baselines we used (FSNet and OneNet) are strictly linked to the use of a specific architecture (TCN).

---

### Author Response · Authors · 2025-12-03

Dear Area Chair, dear reviewers,
We write this final comment to summarize the changes we made to the paper as required by the reviewers. We want to thank everyone one more time for the suggestions and effort made.

### Changes
We have just uploaded a new revised version that we think addresses many of the weaknesses raised by the reviewers.

Following the reviewers' comments, we improved the presentation of the paper. In our work, many themes are explored (time series forecasting, natural gradient, score-driven models, continual learning), and the paper required deep familiarity with all the different subjects. To avoid this, we rewrote multiple parts of our work to provide a more intuitive and high-level overview, added bullet points to explain the main contributions, and clearly annotated the algorithm. We think the paper is significantly clearer now by incorporating many of the reviewers' suggestions.

We also included a full hyperparameter analysis in Appendix D as suggested by reviewer ALYR, and we better explained the reason and plausibility of the bi-Lipschitz condition required for Proposition 4.3, as suggested by reviewer xQJz.


###  Clarifications about the experimental setup
Unfortunately, as explained to the reviewers, online continual learning for forecasting is a very recent field. While it has great potential, the existing alternative methods are still limited to a single architecture (TCN). For this reason, the comparison with different models (transformers, foundation models, ...) would have been impossible to make without changing the nature of other SOTA proposals. Moreover, TCNs are fast to train, which is fundamental in OCL, while preliminary experiments with Transformers showed inconsistent results, suggesting that the Transformers cannot be naively applied in this setting (a conclusion confirmed also in the FSNET paper).


For these reasons, we followed the experimental procedure used in the FSNET and OneNet papers, increasing the number of real-world datasets. On 5 out of 7 datasets, our method performs the best. For transparency, we also included the two datasets where our proposal does not perform well, while proposing a variation of the method for such datasets.

We hence believe that, given the recency of the field, our experimental procedure is complete and fair, improving on current SOTA using the same experimental procedure, while also increasing the number of tested datasets.



### Theoretical contribution
We also want to underline that our paper aims to propose a new optimization approach for time series forecasting, with some strong theoretical grounding and reasonable justification. While we put much of the focus on methodological research, we still obtained an algorithm that, without any architectural solution, is capable of obtaining strong empirical results simply by using smarter optimization steps. We hope that our theoretical and methodological contributions will not be overshadowed by an excessive focus on empirical applications only.

We believe that the forecasting literature could benefit from custom optimizers, and we believe our work provides a first key contribution in this direction.

---

### Meta-Review · Area_Chair_sAkS · 2026-01-05

**Summary:**

The paper studies "online time series forecasting (OTSF)", which  is a real-world problem where data evolve in time and success depends on both rapid adaptation and long-term memory. The authors reframed neural network optimization as a parameter filtering problem, showing that natural gradient descent is a score-driven method; they also studied the formulation from theoretical perspective, and obtained its information-theoretic optimality. In the algorithmic part, they introduced "Natural Score-driven Replay (NatSR)", which combines the robust optimizer with a replay buffer and a dynamic scale heuristic.

The concerns from the reviewers mainly focus on the following points:

1) Limited experiments (Reviewer Pftj, Reviewer ALYR, Reviewer xQJz, Reviewer Nj1d)

2) The theoretical clarity (Reviewer xQJz)

3) Presentation (Reviewer Pftj, Reviewer xQJz)

**Reviewer Concerns:**

The authors provide some explanation to the raised concerns, however, I do feel the current version still has a large room to improve (especially the experimental part). The authors emphasized their main contribution lies in theory, but I find that the real implementation ( Appendix C) contains many relaxations and heuristics. That means, there should be large gap between the proposed theory and the real implementation. In addition, the theoretical analysis shown in Sec. 4 needs more clarification, such as the time and space complexities.

**Reviewer Scores:**

The reviewers give the scores 6, 4, 4, 4. None of them participated in the discussion during rebuttal. I think some of them might slightly increase their scores if they can participate fully in the discussion. Given the high bar of ICLR, I don't think the current version is good enough for acceptance.

---

### Decision · Program_Chairs · 2026-01-26

Reject